# Association between Masseter Muscle Area and Thickness and Outcome after Carotid Endarterectomy: A Retrospective Cohort Study

**DOI:** 10.3390/jcm11113087

**Published:** 2022-05-30

**Authors:** Rianne N. M. Hogenbirk, Louise B. D. Banning, Anita Visser, Harriet Jager-Wittenaar, Robert A. Pol, Clark J. Zeebregts, Joost M. Klaase

**Affiliations:** 1Division of Hepatobiliary Surgery and Liver Transplantation, Department of Surgery, University Medical Center Groningen, University of Groningen, P.O. Box 30.001, 9700 RB Groningen, The Netherlands; r.n.m.hogenbirk@umcg.nl; 2Division of Vascular Surgery, Department of Surgery, University Medical Center Groningen, University of Groningen, P.O. Box 30.001, 9700 RB Groningen, The Netherlands; l.b.d.banning@umcg.nl (L.B.D.B.); r.pol@umcg.nl (R.A.P.); c.j.a.m.zeebregts@umcg.nl (C.J.Z.); 3Department of Oral and Maxillofacial Surgery, University Medical Center Groningen, University of Groningen, P.O. Box 30.001, 9700 RB Groningen, The Netherlands; a.visser@umcg.nl (A.V.); ha.jager@pl.hanze.nl (H.J.-W.); 4Department of Gerodontology, Center for Dentistry and Oral Hygiene, University Medical Center Groningen, University of Groningen, P.O. Box 30.001, 9700 RB Groningen, The Netherlands; 5Research Group Healthy Ageing, Allied Health Care and Nursing, Hanze University of Applied Sciences, P.O. Box 3109, 9701 DC Groningen, The Netherlands

**Keywords:** carotid endarterectomy, computed tomography, frailty, masseter muscle, risk management, sarcopenia

## Abstract

Low muscle mass is associated with adverse outcomes after surgery. This study examined whether facial muscles, such as the masseter muscle, could be used as a proxy for generalized low muscle mass and could be associated with deviant outcomes after carotid endarterectomy (CEA). As a part of the Vascular Ageing study, patients with an available preoperative CT-scan, who underwent an elective CEA between December 2009 and May 2018, were included. Bilateral masseter muscle area and thickness were measured on preoperative CT scans. A masseter muscle area or thickness of one standard deviation below the sex-based mean was considered low masseter muscle area (LMA) or low masseter muscle thickness (LMT). Of the 123 included patients (73.3% men; mean age 68 (9.7) years), 22 (17.9%) patients had LMA, and 18 (14.6%) patients had LMT. A total of 41 (33.3%) patients had a complicated postoperative course and median length of hospital stay was four (4–5) days. Recurrent stroke within 5 years occurred in eight (6.6%) patients. Univariable analysis showed an association between LMA, complications and prolonged hospital stay. LMT was associated with a prolonged hospital stay (OR 8.78 [1.15–66.85]; *p* = 0.036) and recurrent stroke within 5 years (HR 12.40 [1.83–84.09]; *p* = 0.010) in multivariable logistic regression analysis. Masseter muscle might be useful in preoperative risk assessment for adverse short- and long-term postoperative outcomes.

## 1. Introduction

Patients undergoing carotid endarterectomy, a successful treatment for preventing ischemic cerebrovascular accidents, are susceptible to adverse postoperative outcomes, such as a post-procedural stroke, myocardial infarction, and mortality [1]. Preoperative frailty, defined as a state of increased vulnerability due to a decline in physiological reserve and functional capacity of the patient, is known to be associated with such unfavorable outcomes after carotid endarterectomy [2,3,4,5,6]. Although these are separate concepts, the physical component of frailty is sometimes referred to as sarcopenia, and could be diagnosed by measuring (a decline in) skeletal muscle strength and muscle mass [7,8,9,10]. Several studies have measured sarcopenia by using transverse slices of abdominal computed tomography (CT) scans to quantify the total abdominal muscle area and density at the level of the third lumbar vertebra [2,11,12]. Decreased muscle area or muscle density, that is, signs of myosteatosis, were found to be associated with increased adverse outcomes and mortality following surgery [13,14,15,16,17]. CT analysis at the third lumbar vertebra is a useful method of identifying the patients at risk who have a preoperative work up with an abdominal CT scan, but this is often not feasible for patients who are scheduled for non-abdominal surgery, such as head and neck surgery. For these patients, some studies have shown that measuring the masseter muscle area could also be a valid method for evaluating generalized muscle status [18,19]. Oksala et al. found that not only that the masseter muscle area and density could easily and reliably be measured from preoperative CT images, but also that the mean masseter muscle area was associated with postoperative long-term survival in carotid endarterectomy patients [20]. Limited research was performed on masseter muscle measurements and surgical outcomes. Perhaps due to its role as the chewing muscle, the masseter muscle could be of value in assessing nutritional status, or could be identified as a potential proxy for generalized low muscle mass, and, therefore, be used as a risk factor for deviant postoperative outcomes after surgery for patients for whom an abdominal CT scan is absent [21].

The primary aim of this study was to evaluate the association between the masseter muscle area or thickness and adverse short-term and long-term postoperative outcomes after carotid endarterectomy. The secondary aim was to investigate the relationship between masseter muscle area or thickness and frailty, as indicated by the Groningen Frailty Indicator (GFI).

## 2. Materials and Methods

### 2.1. Study Design

This study is part of a large prospective cohort study, initiated in 2009 in our tertiary referral teaching hospital, to study frailty in elective vascular surgery patients (Vascular Ageing Study) [3,22]. Consecutive patients who were scheduled for carotid endarterectomy between December 2009 and May 2018, for whom recent (<12 months) digital preoperative CT and CT angiography (CTA) scans were performed, were included in the study.

The electronic medical records and vascular registry were accessed to obtain information on patient demographics, including sex, age, body mass index (BMI), smoking history, the American Society of Anesthesiologists (ASA) classification, and comorbidities expressed by the Charlson Comorbidity Index. Preoperative screening reports of anesthesiologists were assessed to gather data on the dental status of the patient; data were dichotomized into patients with teeth and into patients who used a partial or full dental prosthesis.

Follow-up and clinical data from the Vascular Ageing Study were used. The Institutional Review Board approved dispensation in accordance with Dutch law on patient-based medical research (WMO) obligations (registration no. METc2016/322). Consequently, informed consent was not obtained. Patient data were processed and electronically stored in agreement with the Declaration of Helsinki’s ethical principles for medical research involving human subjects (2013). Data were stored and analyzed anonymously.

### 2.2. Outcome Parameters

The primary outcome parameters were postoperative surgical outcomes as postoperative complications, length of hospital stay, recurrent stroke beyond 30 days within 5 years, and 5-year survival after CEA. The secondary outcome parameter was preoperative frailty measured by the GFI and its association with masseter muscle measurements. Complications were first classified according to the Clavien–Dindo scoring method, after which the Comprehensive Complication Index (CCI) was calculated [23]. The CCI takes the quantity of appearances of each complication into account, summing all of the postoperative complications, weighted according to their severity, ranging from zero (no complications) to 100 (death) [24]. Length of hospital stay was dichotomized, where the cutoff point for a prolonged hospital stay was defined as a length of hospital stay of seven or more days (≥7 days). Frailty was measured at the out-patient clinic by using the GFI, a frailty measurement tool developed in the Netherlands. The GFI contains 15 dichotomous self-reported items, comprising of physical, cognitive, social, and psychological domains. The frailty score could range from a score of 0 (normal activity without restriction) to 15 (completely disabled) [4,25]. Based on previous publications, a score of ≥4 was considered indicative of frailty [26,27].

### 2.3. Masseter Measurements

Measurements of the masseter muscle area and thickness were performed on preoperative CTs or CTAs of cervical and cerebral vasculature. All of the images were assessed through a picture archiving and communications system (PACS, AGFA Healthcare, Brentford, UK). All of the acquired scans had a slice thickness of 1–3 mm (mm), using a 512 × 512 matrix. Prior to conducting the measurements, adjustments in the tilt alignment of the CT scans were made according to a tangent along the lower borders of the arcus zygomaticus in the coronal plane and along the os palatum nasi in the saggital plane. Next, in the coronal view, the masseter muscle was identified 20 mm below the zygomatic arch. The outlines of the masseter muscle were manually traced bilaterally in the transversal images, and the maximum transversal diameter of the masseter muscles was measured at the same level (Figure 1). The imaging analysis program (PACS, AGFA Healthcare, Brentford, UK) calculated the area in square millimeters (mm^2^) and the masseter thickness in mm. The mean area and thickness, determined by measuring both of the masseter muscles, were used for analysis. In the analyses, low muscle mass, in this study referred to as low masseter muscle area (LMA) or low masseter muscle thickness (LMT), was defined as the mean masseter area or thickness of one standard deviation below the sex-based mean masseter area or the thickness of the study population [28]. Patients with a masseter muscle area or thickness above the cutoff-point of one SD below the sex-based mean were defined as adequate masseter muscle area (AMA) or adequate masseter muscle thickness (AMT).

### 2.4. Statistical Analysis

Categorical variables were compared using Pearson’s χ^2^ test and are presented as numbers and percentages. Continuous variables were analyzed using a Student’s *t*-test (normal distribution) or the Mann–Whitney U test (skewed distribution). Normally distributed variables were presented as mean with standard deviation (SD) and skewed variables as median with interquartile range (IQR). The normality of the data was visually tested using histograms. 

Multivariable logistic regression models were conducted to evaluate the association between the length of hospital stay and LMA and LMT. A multivariable linear regression model was used to assess the association between LMA and LMT and complications by CCI. Based on the existing literature, models were adjusted for the factors which have a strong a priori association with masseter muscle measurements: a complicated postoperative course; length of hospital stay; recurrent stroke or survival. Besides LMA and LMT, factors included in the multivariable linear regression model to assess complications in CCI were sex, age, BMI, ASA score ≥ 3, dental status and current smoking [5,21,29,30]. The multivariable logistic regression analyses to assess the length of hospital stay and LMA or LMT were adjusted for sex, age, BMI, ASA score ≥ 3, dental status, current smoking, and complications in CCI. The results of univariable and multivariable linear regression models were expressed as effect size with 95% confidence interval (95%-CI), and effect sizes of univariable and multivariable logistic regression models were presented as odds ratio (OR) with 95%-CI.

To determine the effect of low masseter muscle area or thickness on patients’ five-year survival and on the occurrence of a recurrent stroke within 5 years, a Cox proportional hazards model, adjusted for sex, age, BMI, ASA score ≥ 3, and dental status, was utilized. A *p* value of <0.05 was considered statistically significant. All of the statistical analyses were performed using the Statistical Package for the Social Sciences (SPSS 23.0, SPSS, Chicago, IL, USA). 

## 3. Results

### 3.1. Demographics

The total cohort of the Vascular Aging study consisted of 1306 patients. In total, 254 patients who had undergone elective carotid endarterectomy were identified. Of these 254 patients, 60 patients had no head or neck CT or CTA available in our hospital, 66 patients had a head CT with insufficient imaging of the masseter muscles to perform muscle measurements, and 5 patients had a CT or CTA that was performed more than twelve months before surgery. Therefore, the remaining 123 patients had a preoperative CT or CTA scan sufficient for measurements and, thereby, formed the basis of the final analysis. Demographics, comorbidities, CCI, and the length of hospital stay did not differ significantly between the included and excluded carotid endarterectomy patients. Patient characteristics and clinical data are summarized in Table 1. The majority of the patients were male (*n* = 90; 73.2%), and the mean (SD) age of the total group was 68 (9.7) years. The majority of patients (*n* = 99 (80.5%) patients) underwent surgery because of a carotid stenosis of minimally 70%. A total of 19 (15.4%) patients had an asymptomatic stenosis prior to surgery, while 17 (13.8%) had a history of amaurosis fugax, 41 (33.3%) had a history of transient ischemic attack(s), and 46 (37.4%) had a history of minor ischemic stroke. 

### 3.2. Masseter Area

The mean (SD) masseter area for males and females was 418 (88) mm^2^ and 344 (87) mm^2^, respectively. Based on the sex-based mean (SD) area of the masseter muscle, the total cohort was divided into a group with low muscle area, the LMA group (*n* = 22), and the AMA group (*n* = 101) (cutoff-points for the LMA group were ≤330 mm^2^ and ≤257 mm^2^ for males and females, respectively). The mean (SD) BMI in the AMA group was 27.6 (4.5) kg/m^2^, which was significantly higher than the mean (SD) BMI of 25.1 (3.1) kg/m^2^ in the LMA group (*p* = 0.014). In the LMA group, a significantly larger proportion of the patients had ASA scores of three or higher (15 (68.1%), versus 45 (44.6%); *p* = 0.037) (Table 1).

### 3.3. Masseter Thickness

The mean (SD) masseter thicknesses for males and females were 12.75 (2.11) mm and 11.54 (2.10) mm, respectively. The total cohort was, based on sex-based mean masseter muscle thickness, divided into LMT (*n* = 18) and AMT (*n* = 105) groups (cutoff-points for the LMT group were ≤10.64 mm and ≤9.44 mm for males and females, respectively). In the LMT group, the mean (SD) BMI was significantly lower in the LMT group compared with the AMT group (24.9 (3.1) versus 27.6 (4.6), *p* = 0.021). In addition, the LMT patients had an ASA score of three or higher significantly more often than the AMT group (83.3% versus 42.9%; *p* = 0.001) (Table 1). A significantly higher proportion of patients with LMT had an ischemic stroke prior to CEA compared with the AMT group (11 (61.1%) versus 35 (33.3%), *p* = 0.024).

### 3.4. Association of Masseter Muscle Area and Thickness with Clinical Outcome Parameters

All of the clinical outcome parameters, including the differences concerning the measured masseter muscle area and thickness, are listed in Table 2. A total of 41 (33.3%) patients had a complicated postoperative course, with nerve injury as the most common occurring complication (15 patients (12.2%)). Patients with LMA more often had postoperative hyper- or hypotension compared with the AMA group (4 (18.2%) versus 5 (5.0%), *p* = 0.031). Although not statistically significant, postoperative stroke occurred more often in the LMA and LMT groups compared with AMA and AMT, respectively (area: 2 (9.1%) versus 4 (4.0%), *p* = 0.311; thickness: 2 (11.1%) versus 4 (3.8%), *p* = 0.184). For the total cohort median (IQR) the CCI was zero (0–8.7), the median (IQR) length of hospital stay was four (4–5) days, and 15 (12.2%) patients had a prolonged hospital stay of seven or more days. The presence of preoperative LMA resulted in a significantly longer median (IQR) length of hospital stay than the AMA group (5.5 (4–7) versus 4.5 (4–5) days; *p* = 0.011). There was also a significant difference in a prolonged hospital length of stay (≥7 days) between the LMA and the AMA groups (six (27.3%) versus nine patients (8.9%), *p* = 0.017). After discharge, no differences were found in 30-day readmission (one (4.5%) versus five (5.0%), *p* = 0.708) or in five-year survival (17 (77.2%) versus 85 (84.2%), *p* = 0.531). For the LMT group, a significantly larger group had a prolonged hospital stay of seven or more days than in the AMT group (5 (27.8%) versus 10 (9.5%), *p* = 0.029). In addition, a significantly larger proportion of the patients with LMT had a recurrent, postoperative ischemic stroke within 5 years (two (11%) versus one (1.9%), *p* = 0.042) (Table 2). 

Univariable linear regression analysis showed that patients with LMA had a significantly increased risk of developing postoperative complications (effect size 5.23 [95%-CI, 0.31–10.14]; *p* = 0.037). After adjusting the model for sex, age, BMI, ASA score ≥ 3, dental status, and current smoking, the association lost statistical significance (4.26 [95%-CI, −1.26 to 9.79]; *p* = 0.129). Moreover, no significant association between masseter muscle thickness and complications was observed in the multivariable linear regression analysis (1.15 [95%-CI, −4.99 to 7.29]; *p* = 0.712). In the multivariable logistic regression analysis, a significantly increased risk of prolonged hospital length of stay was observed in patients with LMT (OR 8.78 [95%-CI, 1.15–66.85; *p* = 0.036). This increased risk was also observed in the multivariable analysis for the LMA group (OR 4.73 [95%-CI, 1.13–19.73]; *p* = 0.033). Regarding long-term postoperative outcome, using multivariable cox-regression analysis, in patients with low masseter muscle thickness a significantly increased postoperative risk was observed for recurrent stroke within 5-years (HR 12.40 [95%-CI, 1.83–84.09]; *p* = 0.010) (Table 3). 

### 3.5. Association of Masseter Muscle Area and Thickness with GFI

All of the included 123 patients underwent both frailty and masseter muscle assessment, of whom a total of 34 (27.6%) patients scored four or higher on the GFI. No differences were found in GFI > 3 between the LMA group and AMA group (6 (27.3%) versus 28 patients (27.7%); *p* = 0.966). Neither difference was found between the LMT group and AMT group (6 (33.3%) versus 28 patients (26.7%); *p* = 0.559).

## 4. Discussion

In this study, preoperative CT-scan-based masseter muscle area and thickness were associated with adverse surgical outcomes after carotid endarterectomy. Low masseter muscle thickness was independently associated with a prolonged hospital stay (≥7 days) and recurrent postoperative ischemic stroke within 5 years. No significant association between masseter muscle area or thickness and GFI was observed.

The association between the preoperative muscle quantity and quality and surgical outcomes in oncological and other abdominal surgical patients was previously studied extensively by using total abdominal muscle area [2,3,12,13,14,15], but only a few studies have shown that measuring the masseter muscle area could also be a valid alternative method for evaluating generalized muscle status. Wallace et al. reported a reliable approach to measure the masseter muscle with a significant correlation between the masseter and psoas muscles in trauma patients [18]. In addition, several studies have validated reproducibility and the independence of observer bias in measuring the masseter muscle area two centimeters below the arcus zygomaticus on CT scans [18,20,28,31]. Yamaguchi et al. showed that, in an elderly population residing in Japan, masseter muscle thickness, as measured by ultrasound, was significantly correlated to hand grip strength and walking speed, and therefore could be a possible method of determining physical frailty [19]. However, in the present study, although frailty and sarcopenia are frequently overlapping concepts, no statistically significant association was found between a higher GFI and LMA or LMT. Possibly, as stated before, the concept of frailty consists of multiple physical and physiological factors to indicate a state of increased vulnerability, while the presence of low muscle mass only reflects one part of the physical domain of the concept of frailty [8,9,10]. 

Some studies have focused on patient demographics in relation to the masseter muscle area, but only a few have focused on the association between masseter muscle measurements and surgical outcomes. A summary of the existing literature on masseter muscle measures and surgical outcomes is shown in Table 4. Since this summary was not conducted as a systematic review, there is a potential for publication bias in this overview. Nonetheless, an association between the masseter muscle area and short- and mid-term patient survival was found in trauma patients [18,28]. In addition, patients with severe traumatic brain injury and masseter sarcopenia, as defined by a masseter muscle area below one standard deviation from the sex-based mean, were less often discharged to home, but more often discharged to other healthcare facilities [28]. Oksala et al. studied the masseter muscle area and long-term surgical outcomes after carotid endarterectomy [20]. However, in contrast to the research by Oksala et al., in our study, LMA was not a significant predictor of diminished long-term patient survival after carotid endarterectomy. The smaller sample size and low mortality rate in our study probably resulted in an inability to detect a significant association between the masseter muscle area and survival.

However, despite the low numbers, our multivariable Cox regression analysis showed a significantly increased hazard ratio for recurrent stroke within 5 years in patients with LMT. In addition, compared to previous reports on short-term surgical outcomes, our analysis showed that LMA was significantly associated with a higher CCI and a longer hospital stay. After adjusting the model for sex, age, BMI, ASA, and complications, both low masseter thickness and low masseter area remained significantly associated with a prolonged hospital length of stay. The predictive value of masseter muscle thickness on the short-term postoperative length of hospital stay and long-term recurrent stroke risk within 5 years could be of value for future prospective research. Due to the superficial anatomical position of the masseter muscle, thickness could also be assessed by using point of care ultrasound (POCUS) [19]. POCUS could be used as a simple bedside tool to assess the masseter muscle thickness, and it could be a useful method to screen for low muscle mass in patients without a CT scan [32,33]. However, although POCUS could be used as a simple bedside tool, it could be considered as more cumbersome to use in the daily clinical practice, compared with BMI and frailty indices. Nonetheless, as beforementioned, frailty indices consist of multiple domains, whereas in our study the masseter muscle thickness was not correlated with these frailty indices. Thereby, regarding BMI, in a recent study conducted by Arinze et al., ‘underweight’, defined as BMI < 18.5 kg/m^2^, was associated with dismal short- and long-term survival [34]. However, in our study, although lower BMI was associated with LMA and LMT, the mean BMI in these groups would be, in the study of Arinze et al., classified as ‘normal weight’ (18.5–24.2 kg/m^2^), and would not be considered as a risk for dismal surgical outcomes. Therefore, in our opinion, masseter muscle measurement could be of added value to use as an additional tool in preoperative risk-assessment, next to existing factors such as frailty indices and BMI.

The results of this study suggest that preoperatively measured masseter muscle size could be used as a clinical marker to identify vulnerable patients and to contribute to individualized preoperative patient care by assessing nutritional status and intensifying dietary and physical support. However, a note of caution is due here because of the uncertainty in the literature regarding associations of sarcopenia with outcomes among vascular patients. For example, the meta-analysis by Houghton et al. was not able to find an association between CT-assessed sarcopenia and surgical outcomes, while another meta-analysis by Antoniou et al. found an association. Nonetheless, optimizing a patient’s physical fitness or nutritional status before a vascular intervention may improve outcomes [35,36]. For example, in patients undergoing elective abdominal aortic aneurysm repair, pre-habilitation, with a supervised exercise program six weeks before the operation, led to fewer postoperative complications and a shorter length of hospital stay [37]. However, in carotid surgery, according to vascular surgical guidelines, the interval between the (transient) ischemic cerebral vascular event and the intervention should be less than two weeks, to minimize the risk of a fatal or disabling stroke [38,39]. The consequences of a possible recurrent event after delayed treatment outweigh the adverse short-term surgical outcome of a prolonged hospital stay. Future research should, therefore, focus on the rapid improvement of preoperative physical conditions after identifying vulnerability by masseter muscle measurements, or on incorporation of masseter muscle measurements into outpatient follow-ups, when asymptomatic carotid stenosis is detected.

Several limitations of our study need to be addressed. First, limited research is available on the mean masseter muscle area and thickness for a given age and sex; therefore, no validated truncation value was available to compare both groups. Second, it is important to acknowledge that the study population was small, since a large group of the patients undergoing carotid endarterectomy had no suitable CT imaging to enable masseter muscle measurement. Therefore, our study may be underpowered to make a risk-prediction model and to detect any significant differences in postoperative surgical outcome and survival. Third, due to the retrospective design of our study, CT scans used for measurements of the masseter muscle were performed up to 12 months before surgery, which could have caused either an underestimation or an overestimation of the masseter muscle area or thickness. In addition, no standardized contrast scan protocol was utilized; thus, the effect of contrast enhancement on muscle composition varied and, therefore, the masseter muscle density measures from scan protocols could not be reliably compared [40]. Another limitation, caused by the retrospective design of the study, is the fact that we were not able to perform a dental assessment by a dentist. By using anesthesiologists’ reports, we were only able to determine the use of a dental prosthesis. However, no full assessment of the chewing ability, oral health, and functionality of the teeth and denture factors could be included in our analyses. Lastly, due to the tertiary referral nature of our clinic, the patients in this study were somewhat less generalizable than other cohorts of vascular surgical patients, therefore, in future research a validation cohort derived from multiple peripheral hospitals should be collected. 

## 5. Conclusions

Only limited research was conducted on the masseter muscle area or thickness and post-surgical outcomes. Although not associated with the GFI, the results of this study identify an important relationship between the masseter muscle thickness and short- and long-term surgical outcomes after carotid endarterectomy, thereby highlighting the potential usefulness of the masseter muscle for risk stratification in surgical vascular patients. Masseter muscle measurement could contribute to, but not determine, individualized health care. Future prospective research should focus on measuring the masseter muscle area and thickness, both by using CT scans and by using POCUS, in different fields of surgery.

## Figures and Tables

**Figure 1 jcm-11-03087-f001:**
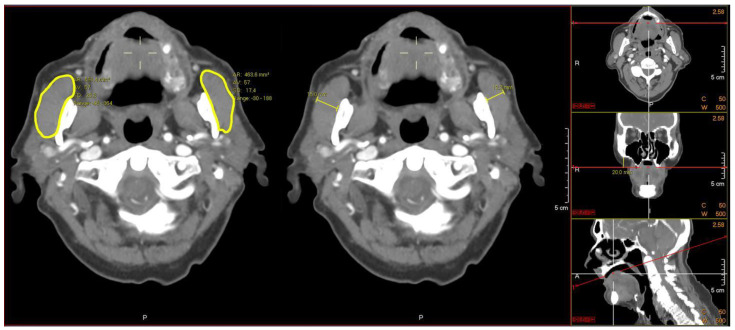
Example of performed measurements of bilateral masseter muscle area (**left**) and thickness (**middle**) after tilt adjustments were performed (**right**).

**Table 1 jcm-11-03087-t001:** Comparison of clinical characteristics by presence of adequate or low masseter muscle area and adequate or low masseter muscle thickness.

Demographics		Masseter Muscle Area			Masseter Muscle Thickness		
	All Patients (*n* = 123)	AMA (*n* = 101)	LMA (*n* = 22)	*p* Value	AMT (*n* = 105)	LMT (*n* = 18)	*p* Value
Sex							
Male	90 (73.2%)	74 (73.3%)	16 (72.7%)	0.959	76 (72.4%)	14 (77.8%)	0.633
Female	33 (26.8%)	27 (26.7%)	6 (27.3%)	0.959	29 (27.6%)	4 (22.2%)	0.633
Age—y	68 (9.7)	68 (9.9)	71 (8.7)	0.652	68 (10.3)	69 (6.1)	0.475
BMI—kg/m^2^	27.2 (4.5)	27.6 (4.5)	25.1 (3.1)	0.014	27.6 (4.6)	24.9 (3.1)	0.021
Smoking status							
Current	33 (26.8%)	24 (23.8%)	9 (40.9%)	0.087	23 (21.9%)	10 (55.6%)	0.005
History	38 (30.9%)	35 (34.7%)	3 (13.6%)	0.041	34 (32.4%)	4 (22.2%)	0.389
Never	52 (42.3%)	42 (41.6%)	10 (45.5%)	0.814	48 (45.7%)	4 (22.2%)	0.051
ASA ≥ 3	60 (48.7%)	45 (44.6%)	15 (68.1%)	0.037	45 (42.9%)	15 (83.3%)	0.001
Charlson Comorbidity Index—median (IQR)	5 (3–6)	5 (3–6)	5 (4–5)	0.806	4 (3–6)	5 (4–5)	0.496
Partial or full dental prosthesis	65 (52.8%)	54 (53.5%)	11 (50.0%)	0.446	53 (50.5%)	12 (66.7%)	0.234
GFI	2 (1–4)	2 (1–4)	2 (1–5)	0.827	2 (1–4)	2 (1–5.25)	0.233
GFI > 3	34 (27.6%)	28 (27.7%)	6 (27.3%)	0.966	28 (26.7%)	6 (33.3%)	0.559
Indication							
Asymptomatic	19 (15.4%)	16 (15.8%)	3 (13.6%)	0.795	18 (17.1%)	1 (5.6%)	0.209
Amaurosis	17 (13.8%)	16 (15.8%)	1 (4.5%)	0.164	15 (14.3%)	2 (11.1%)	0.718
Transient ischemic attack	41 (33.3%)	33 (32.7%)	8 (36.4%)	0.739	37 (35.2%)	4 (22.2%)	0.279
Minor ischemic stroke	46 (37.4%)	36 (35.6%)	10 (45.5%)	0.389	35 (33.3%)	11 (61.1%)	0.024
Ipsilateral stenosis (%)							
<50%	2 (1.6%)	1 (1.0%)	1 (4.5%)	0.232	1 (1%)	1 (5.6%)	0.154
50–70%	22 (17.9%)	17 (16.8%)	5 (22.7%)	0.513	22 (21%)	0 (0%)	0.032
>70%	99 (80.5%)	83 (82.2%)	16 (72.7%)	0.311	82 (78%)	17 (94.4%)	0.106

Low muscle area and thickness were defined by a mean masseter muscle area or thickness of one standard deviation below the sex-based mean. Data are presented as mean (SD) or median (IQR), unless indicated otherwise. AMA: adequate masseter muscle area; LMA: low masseter muscle area; AMT: adequate masseter muscle thickness; LMT: low masseter muscle thickness; BMI: body mass index; ASA: American Society of Anesthesiologists’ score; GFI: Groningen Frailty Indicator.

**Table 2 jcm-11-03087-t002:** Comparison of clinical outcomes by presence of adequate or low masseter muscle area and adequate or low masseter muscle thickness.

Clinical Outcomes		Masseter Muscle Area			Masseter Muscle Thickness		
	All Patients (*n =* 123)	AMA (*n =* 101)	LMA (*n =* 22)	*p* Value	AMT (*n =* 105)	LMT (*n =* 18)	*p* Value
Complications within 30 days	41 (33.3%)	30 (29.7)	11 (50.0%)	0.067	34 (32.4%)	7 (38.9%)	0.588
Cardiac complication	6 (4.9%)	4 (4.1%)	2 (9.1%)	0.311	1 (5.0%)	1 (5.6%)	0.885
Hyper/hypotension	9 (7.3%)	5 (5.0%)	4 (18.2%)	0.031	6 (5.7%)	3 (16.7%)	0.099
Stroke	6 (4.9%)	4 (4.0%)	2 (9.1%)	0.311	4 (3.8%)	2 (11.1%)	0.184
Wound	8 (6.5%)	7 (6.9%)	1 (4.5%)	0.681	7 (6.7%)	1 (5.6%)	0.860
Nerve injury	15 (12.2%)	13 (12.9%)	2 (9.1%)	0.623	15 (14.3%)	0 (0%)	0.087
Other	3 (2.4%)	2 (2.0%)	1 (4.5%)	0.480	3 (2.9%)	0 (0%)	0.468
Complications in CCI, median (IQR)	0 (0–8.7)	0 (0–8.7)	0 (0–20.9)	0.082	0 (0–8.7)	0 (0–20.9)	0.624
Length of hospital stay (days), median (IQR)	4 (4–5)	4.50 (4–5)	5.5 (4–7)	0.011	4 (4–5)	4 (4–7.25)	0.621
Hospital stay ≥ 7 days	15 (12.2%)	9 (8.9%)	6 (27.3%)	0.017	10 (9.5%)	5 (27.8%)	0.029
Readmission within 30 days	6 (4.9%)	5 (5.0%)	1 (4.5%)	0.708	5 (4.8%)	1 (5.6%)	0.62
Recurrent stroke within 5 years (beyond 30 days)	8 (6.6%)	6 (5.9%)	2 (9.1%)	0.587	5 (4.8%)	3 (16.7%)	0.058
Transient ischemic attack	4 (3.3%)	3 (3.0%)	1 (4.5%)	0.706	3 (2.9%)	1 (5.6%)	0.551
Ischemic stroke	4 (3.3%)	3 (3.0%)	1 (4.5%)	0.706	2 (1.9%)	2 (11.1%)	0.042
5-year survival	102 (83.9%)	85 (84.2%)	17 (77.2%)	0.531	88 (83.8%)	14 (77.8%)	0.368

Low muscle area and thickness were defined by a mean masseter muscle area or thickness of one standard deviation below the sex-based mean. Data are presented as mean (SD) or median (IQR), unless indicated otherwise. AMA: adequate masseter muscle area; LMA: low masseter muscle area; AMT: adequate masseter muscle thickness; LMT: low masseter muscle thickness; CCI: Comprehensive Complication Index

**Table 3 jcm-11-03087-t003:** Univariable and multivariable linear and logistic regression analysis of the association between low masseter muscle area, low masseter muscle thickness, and surgical outcome.

	Low Masseter Muscle Area				Low Masseter Muscle Thickness			
Outcome Parameter	Effect Size	95%-CI		*p* Value	Effect Size	95%-CI		*p* Value
		Lower	Upper			Lower	Upper	
Complications (CCI)								
Univariable analysis	5.23	0.31	10.14	0.037	2.92	−2.48	8.32	0.287
Multivariable analysis *	4.26	−1.26	9.79	0.129	1.15	−4.99	7.29	0.712
Hospital stay ≥7 days								
Univariable analysis	OR 3.80	1.20	12.24	0.023	OR 3.65	1.08	12.38	0.037
Multivariable analysis **	OR 4.73	1.13	19.73	0.033	OR 8.78	1.15	66.85	0.036
5-year recurrent stroke risk (beyond 30 days)								
Univariable analysis	HR 1.70	0.34	8.4	0.518	HR 4.04	0.96	16.93	0.056
Multivariable analysis *	HR 2.05	0.29	14.62	0.473	HR 12.40	1.83	84.09	0.010
5-year survival								
Univariable analysis	HR 1.56	0.58	4.31	0.374	HR 1.54	0.52	4.59	0.436
Multivariable analysis **	HR 1.25	0.38	4.14	0.714	HR 1.63	0.47	5.72	0.443

Low masseter muscle area and thickness were defined by a mean masseter muscle area of 1 standard deviation below the sex-based mean. * adjusted for: Sex; age; BMI; ASA score ≥ 3; dental status; current smoking. ** adjusted for: Sex; age; BMI; ASA score ≥ 3; dental status; current smoking; complications in CCI. CCI: Comprehensive Complication Index.

**Table 4 jcm-11-03087-t004:** An overview of existing literature regarding CT-based masseter measurements and surgical outcome.

Author (Year) Country [Citation]	Population	Sample	Masseter Measurements		Results	
			Type of Measurements	Mean	Outcome Parameters	Effect Size [95%-CI]
Lindström et al. (2021) Finland [41]	Pre-operative before mechanical thrombectomy in acute ischemic stroke patients	312	Masseter area	M 450 mm^2^ F 360 mm^2^	3-month survival	OR 0.57 [0.35–0.91] *p* = 0.02
			Masseter density	M 65.5 HUF 61 HU	3-month survival	OR 0.61 [0.41–0.92] *p* = 0.02
Waduud et al. (2020) United Kingdom [31]	Pre-operative before carotid endarterectomy	149	Masseter area	-	30-day survival	HR 0.14 [0–5.44] *p* = 0.29
					1-year survival	HR 0.51 [0.13–2.07] *p* = 0.35
					4-year survival	HR 0.36 [0.13–1.04] *p* = 0.06
					Overall survival	HR 0.38 [0.15–0.97] *p* = 0.04
			Masseter density	-	30-day survival	HR 1.03 [0.95–1.10] *p* = 0.52
					1-year survival	HR 0.98 [0.95–1.01] *p* = 0.31
					4-year survival	HR 0.98 [0.96–1.01] *p* = 0.24
					Overall survival	HR 0.97 [0.95–1.00] *p* = 0.11
			Masseter area × masseter density	-	30-day survival	HR 0.99 [0.97–1.02] *p* = 0.57
					1-year survival	HR 0.99 [0.98–1.00] *p* = 0.20
					4-year survival	HR 0.99 [0.98–1.00] *p* = 0.07
					Overall survival	HR 0.99 [0.99–1.00] *p* = 0.04
Tanabe et al. (2019) United States of America [42]	Geriatric trauma patients (≥65 years) admitted to the traumatic ICU	327	Masseter area	M 439 mm^2^ F 348 mm^2^	1-year mortality	HR 2 [1.20–3.10] *p* = 0.005
Oksala et al. (2019) Japan [20]	Pre-operative before carotid endarterectomy	242	Masseter area	M 420 mm^2^ F 349 mm^2^	Long-term survival—Median follow-up 68 months	HR 0.75 [0.58–0.97] *p* = 0.03
			Masseter density	M 55 HU F 49 HU	Long-term survival—Median follow up 68 months	HR 0.92 [0.76–1.12] *p* = 0.43
Hu et al. (2018) United States of America [28]	Geriatric trauma patients (≥55 years) with sTBI	108	Masseter area	M 455 mm^2^ F 337 mm^2^	30-day mortality	HR 0.78 [0.62–0.97] *p* = 0.04
			Masseter area 1SD below sex-based mean	-	Discharge disposition to home	0% vs. 13%, *p* = 0.04
					30-day mortality	OR 2.95 [1.03–8.49] *p* = 0.05
Wallace et al. (2016) United States of America [18]	Blunt-injured trauma geriatric patients (≥65 years)	357	Masseter area	M 418 mm^2^ F 343 mm^2^	2-year mortality	HR 0.76 [0.60–0.96] *p* = 0.02

Six studies have reported the association between CT-measured Masseter area or density and outcome after trauma or surgery. Data regarding mean masseter measurements were listed where available, and main conclusions of the studies are reported. M: male; F: female; sTBI: severe traumatic brain injury; HU = Hounsfield units; ICU: Intensive Care Unit.

## Data Availability

Anonymized data that support the findings of this study will be made available upon request to the corresponding authors after the date of publication and after completion of a data sharing agreement.

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
