# Peer review of "Association between Masseter Muscle Area and Thickness and Outcome after Carotid Endarterectomy: A Retrospective Cohort Study"

_jcm, 2022, doi:10.3390/jcm11113087_

Round 1

Reviewer 1 Report

A review for Journal of Clinical Medicine:

Re “ Association between masseter muscle area and thickness and outcome after carotid endarterectomy: a retrospective cohort study”

This is an interesting study. However, I have some major concerns regarding the statical modelling approach alongside other minor comments.

Abstract:

-L24: there is no need to report the frequency of males, the percentage is enough. Remove the frequency, please.

L 90-91: not clear, which are the secondary outcomes.

“Secondary outcome parameters included the association between masseter muscle area and thickness and preoperatively measured frailty. “

Methods:

How was dental status examined?

The authors should add a section to describe how were the covariates examined in the study.

“Multivariable logistic regression models were conducted to evaluate the association between length of hospital stay and LMA and LMT. A multivariable linear regression model was used to assess LMA and LMT and complications by CCI. Models were adjusted for confounding variables, which in univariable analysis were nominally associated (P<.1) with masseter muscle measurements or surgical outcome. Based on existing literature, variables assessed for univariable analysis were sex, age, BMI, smoking status, ASA score, and dental status. “

The modelling strategy is confusing. Selecting only statistically significant variables from univariable analysis (Univariable prefiltering) is a problematic practice that might cause ruling out some important adjustment variables needed for control in the model such as age in this study. See for instance,

Sun, G.W., Shook, T.L. and Kay, G.L., 1996. Inappropriate use of bivariable analysis to screen risk factors for use in multivariable analysis. Journal of clinical epidemiology49(8), pp.907-916.

instead, authors should first draw assumptions about the relationships of variables based on the available evidence and what is known in the literature.

Reviewer 2 Report

Thank you for this paper. The purpose of the study to be found in this study is well described, and I think the direction of the study is also good. This study has a great potential to bring something very useful.

However, I have one question in the research methodology section.

The subjects of this study seem to have a lot of differences in the ratio between male and female. Of course, there was no significant difference between the ratio of male and female(73.2 vs 26.8) and the ratio of male and female according to MMA(73.3, 72.7 vs 26.7, 27.3) and MMT(72.4, 77.8 vs 27.6, 22.2) (p=.959, .633). However, the difference in the ratio of male and female in the study subjects themselves is large (73.2 vs. 26.8). Therefore, it seems that the regression analysis used in Table 3 should also be adjusted for sex variables. I ask for the author's opinion on this.

Also, please modify the measurement unit in the method part line 172~.

(ex. - mm2, kg/m2)

Round 2

Reviewer 1 Report

The authors addressed all my comments and the manuscript has significantly improved. However, the authors should address the way dental status was defined in the study among the study limitations, as the used variable does not indicate the functionality of the teeth and denture factors (quality-use -fit).

Author Response

Reviewer #1:
The authors addressed all my comments and the manuscript has significantly improved. However, the authors should address the way dental status was defined in the study among the study limitations, as the used variable does not indicate the functionality of the teeth and denture factors (quality-use -fit).

Authors reply: We agree with the reviewer that, by only using anesthesiologists reports, we were not able to fully assess oral health and functionality of teeth and dentures. Therefore, we added this as a limitation in our Discussion section (L336-340).